# A New Analysis of Real-Time Fatality Rate in the Initial Stage of COVID-19

**DOI:** 10.3390/e25071028

**Published:** 2023-07-06

**Authors:** Chuanbo Zhou, Jiaohong Fang, Mingzhi Mao

**Affiliations:** School of Mathematics and Physics, China University of Geosciences, Wuhan 430074, China; slxy@cug.edu.cn (C.Z.); fjh_study@163.com (J.F.)

**Keywords:** real-time fatality rate, permutation test, variance analysis, multinomial distribution, 62P10, 62L10, 68P30

## Abstract

Mortality is one of the most important epidemiological measures and a key indicator of the effectiveness of potential treatments or interventions. In this paper, a permutation test method of variance analysis is proposed to test the null hypothesis that the real-time fatality rates of multiple groups were equal during the epidemic period. In light of large-scale simulation studies, the proposed test method can accurately identify the differences between different groups and display satisfactory performance. We apply the proposed method to the real dataset of the COVID-19 epidemic in mainland China (excluding Hubei), Hubei Province (excluding Wuhan), and Wuhan from 31 January 2020 to 30 March 2020. By comparing the differences in the disease severity for differential cities, we show that the severity of the early disease of COVID-19 may be related to the effectiveness of interventions and the improvement in medical resources.

## 1. Introduction

An epidemic is a disease for which observation value exceeds the expected value. Once an epidemic affects a large population or spreads globally, it is called a pandemic [1]. In recent decades, the incidence and mortality of emerging infectious diseases have increased worldwide, causing significant social, political, and economic damage [2]. For example, several large-scale epidemics that have broken out in recent years include Severe Acute Respiratory Syndrome (SARS), H1N1 influenza, and the ebola virus, which have undoubtedly had a huge impact on the social, political, economic, and other aspects of the infected countries. In 2019, the initial outbreak of COVID-19 in Wuhan, Hubei Province, China, spread rapidly around the world, posing a major threat to global public health. According to the World Health Organization [3], through 1 January 2023, more than 656 million confirmed cases of COVID-19 and more than 6.6 million deaths have been reported worldwide.

Mortality is an important epidemiological measurement, which is used to indicate the severity of the disease and measure the virulence of the disease. The World Health Organization has defined mortality as the ratio of the cumulative number of deaths to the cumulative number of confirmed cases. Mortality is used as an indicator to measure the severity of the disease [4]. Yip et al. [5] found that, in the early stage of the epidemic, because the infection was not over, few patients died of the disease, so the estimated fatality rate was low. With the development of the epidemic, more and more people died at the end of the disease, while more and more people were diagnosed, so the estimated rate was still low. Last [6] argued that the simple estimator is not sensitive to changes in mortality during the epidemic and will only perform well at the end of the epidemic.

Lam et al. [7] believed that the mortality of emerging epidemics may change over time, and the downward trend of mortality may reflect the effectiveness of government interventions and the improvement of medical resources. Ambreen et al. [8] considered the impact of meteorological factors on COVID-19. Chen et al. [9] also believed that pairing assistance is an effective way to curb COVID-19. During the outbreak of an epidemic, Lam et al. [7] recommended using real-time fatality rate to measure the severity of the epidemic, rather than traditional mortality. The real-time fatality rate is defined as the probability of death based on the counting process method conditional on death or recovery. Compared with traditional mortality, the real-time fatality rate has been proven to be more sensitive to capturing changes in mortality during the course of an epidemic. Reich et al. [10] defined the relative fatality rate as the ratio of the fatality rate of a group to that of another reference group and compared the fatality rate of a specific group using a generalized linear model framework. Chen et al. [11] used the generalized linear model framework to estimate the fatality rate according to the maximal likelihood method. However, the assumptions about the real-time fatality rate in their methods are very strict and may be more suitable for chronic diseases than for emerging infectious diseases. In order to detect changes in the real-time fatality rate, Yip et al. [12] proposed a competitive risk model implemented by the counting process to estimate the real-time fatality rate. Yip et al. [5] considered the chain multinomial model to estimate the real-time fatality rate. Lam et al. [7] developed a single-sample sequential test for the null hypothesis of constant fatality rate. Qu et al. [13] developed a multi-sample sequential test for the null hypothesis of real-time fatality rate.

During the highly infectious epidemic outbreak, the government needs to formulate effective interventions as soon as possible to effectively curb the spread of the disease to reduce mortality, minimize the severity of the disease, and ensure the safety of people’s lives. Due to various reasons, it is difficult to collect complete and detailed data. However, it is relatively easy to obtain summary data on confirmed cases, deaths, and recoveries of COVID-19 from the official channels of health departments in various regions.

In order to solve the above problem, it is necessary to carry out relevant statistical tests and compare the severity of diseases between multiple groups according to a simple data structure. Inspired by Yip et al. [12], we propose the analysis of variance using permutation test for real-time fatality rates among different groups over a fixed time period. With the accumulation of statistical evidence, the null hypothesis can be rejected. Under the proposed testing procedure, we can also test the differences in real-time fatality rate between different regions, different treatment groups, and different age groups, so as to provide information for the health sector. We recall that Zhao et al. [14] tested the clinical symptoms of patients using Remdesivir and other therapeutic drugs through survival analysis of log-rank and found that the survival rate of patients using Remdesivir was significantly improved within 8 weeks. In essence, the potential use of multi-group trials is to compare the changes in the real-time fatality rate with Remdesivir and standard treatment. In the empirical analysis section, we compare the differences in real-time fatality rate among different regions. The results suggest that the effectiveness of interventions and medical resource assistance implemented by government departments during public health emergencies are useful and critical.

In Section 2, the test statistic is put forward to study the asymptotic distribution. In Section 3, the relevant knowledge of the permutation test is introduced. In Section 4, simulation experiments are carried out to evaluate the effectiveness of the proposed test method in different scenarios. In Section 5, the proposed test procedure is applied to the comparison of disease severity among three independent clusters during the outbreak: mainland China (excluding Hubei Province), Hubei Province (excluding Wuhan), and Wuhan. In Section 6, we summarize the provided results.

## 2. Preliminaries

### 2.1. Test Statistic

In this paper, days are taken as the unit, and time 0 can be set as the day where observation is implemented for a particular group. Time [0,t] is divided into *T* fixed intervals, the research objects are divided into *k* groups according to certain rules, and the sample size of each group is $m_{i}$(i=1,2,…,k). Information about the number of cases, deaths, and recoveries in the group at the end of each interval is collected in order from T=0. Denote the numbers of deaths, recoveries, and hospitalizations in the *i*-th group at the end of day *t* by $n_{i,D}\left(t\right)$, $n_{i,R}\left(t\right)$, and $n_{i,H}\left(t\right)$, respectively. We further denote the cumulative numbers of deaths and recoveries in the group at the end of day *t* by $N_{i,D}\left(t\right)$ and $N_{i,R}\left(t\right)$.

Assuming that the number of confirmed cases in the *i*-th group at the end of day t−1 is $H_{i}\left(t-1\right)$, they are independently and identically distributed for each confirmed patient and will be in one of the following three states on day *t*: death, recovery, hospitalization. That is,
$$H_{i}\left(t-1\right)=n_{i,D}\left(t\right)+n_{i,R}\left(t\right)+n_{i,H}\left(t\right),$$
assuming that the probability of death is $p_{i,D}\left(t\right)$, the probability of recovery is $p_{i,R}\left(t\right)$, and the probability of hospitalization is $p_{i,H}\left(t\right)$. According to the multinomial distribution proposed by Yip et al. [5], we have
(1)$$\begin{array}{cc} \; & P\left(X_{1}=n_{i,D}\left(t\right),X_{2}=n_{i,R}\left(t\right),X_{3}=n_{i,H}\left(t\right)\right)\\ = & \frac{H_{i}\left(t-1\right)!}{n_{i,D}\left(t\right)!n_{i,R}\left(t\right)!n_{i,H}\left(t\right)!}p_{i,D}{\left(t\right)}^{n_{i,D}\left(t\right)}p_{i,R}{\left(t\right)}^{n_{i,R}\left(t\right)}p_{i,H}{\left(t\right)}^{n_{i,H}\left(t\right)}. \end{array}$$

According to Yip et al. [12], for group *i*, death and recovery are regarded as two competing risks. To consider the real-time fatality rate on day *t*, the real-time fatality rate is defined as
(2)$$p_{i}\left(t\right)=\frac{p_{i,D}\left(t\right)}{p_{i,D}\left(t\right)+p_{i,R}\left(t\right)},\;$$
and the kernel function of the probability density function of Equation (Equation 1) is
(3)$$H\left(p\right)=p_{i,D}{\left(t\right)}^{n_{i,D}\left(t\right)}p_{i,R}{\left(t\right)}^{n_{i,R}\left(t\right)}p_{i,H}{\left(t\right)}^{n_{i,H}\left(t\right)},\;$$
therefore the logarithmic likelihood function of Equation (Equation 1) is
(4)$$\begin{array}{cc} l(p)= & \log H(p)\\ = & n_{i,D}\left(t\right)\log p_{i,D}\left(t\right)+n_{i,R}\left(t\right)\log p_{i,R}\left(t\right)+n_{i,H}\left(t\right)\log p_{i,H}\left(t\right). \end{array}$$

For $p_{i,H}\left(t\right)=1-p_{i,D}\left(t\right)-p_{i,R}\left(t\right)$, set $x_{1}=p_{i,D}\left(t\right)$, $x_{2}=p_{i,H}\left(t\right)$, and $p_{i,H}\left(t\right)$ to derivation of $x_{1}$ and $x_{2}$, respectively. We have $\frac{\partial p_{i,H}\left(t\right)}{\partial x_{1}}=\frac{\partial p_{i,H}\left(t\right)}{\partial x_{2}}=-1$, that is,
$$\frac{\partial p_{i,H}\left(t\right)}{\partial p_{i,D}\left(t\right)}=\frac{\partial p_{i,H}\left(t\right)}{\partial p_{i,R}\left(t\right)}=-1,$$
so
$$\frac{\partial \log p_{i,H}\left(t\right)}{\partial p_{i,D}\left(t\right)}=\frac{\partial \log p_{i,H}\left(t\right)}{\partial p_{i,R}\left(t\right)}=-\frac{1}{p_{i,H}\left(t\right)}.$$

Removing the redundancy term, we can regard l(p) as a function of $p_{i,D}\left(t\right)+p_{i,R}\left(t\right)$, with derivations of $p_{i,D}\left(t\right)$ and $p_{i,R}\left(t\right)$ respectively on l(p), and obtain the following likelihood equation:(5)$$\frac{\partial l(p)}{\partial p_{i,D}\left(t\right)}=\frac{n_{i,D}\left(t\right)}{p_{i,D}\left(t\right)}-\frac{n_{i,H}\left(t\right)}{p_{i,H}\left(t\right)},\;$$
(6)$$\frac{\partial l(p)}{\partial p_{i,R}\left(t\right)}=\frac{n_{i,R}\left(t\right)}{p_{i,R}\left(t\right)}-\frac{n_{i,H}\left(t\right)}{p_{i,H}\left(t\right)}.\;$$

Let Equations (5) and (6) be equal to 0; their maximum likelihood estimation solution satisfies ${\hat{p}}_{i,D}\left(t\right)=\frac{{\hat{p}}_{i,H}\left(t\right)}{n_{i,H}\left(t\right)}n_{i,D}\left(t\right)$ and ${\hat{p}}_{i,R}\left(t\right)=\frac{{\hat{p}}_{i,H}\left(t\right)}{n_{i,H}\left(t\right)}n_{i,R}\left(t\right)$, respectively. Since ${\hat{p}}_{i,D}\left(t\right)+{\hat{p}}_{i,R}\left(t\right)+{\hat{p}}_{i,H}\left(t\right)=1$, the maximum likelihood estimation value of $p_{i,D}\left(t\right)$ and $p_{i,R}\left(t\right)$ is
$${\hat{p}}_{i,D}\left(t\right)=\frac{n_{i,D}\left(t\right)}{H_{i}\left(t-1\right)},\;\;\;\;{\hat{p}}_{i,R}\left(t\right)=\frac{n_{i,R}\left(t\right)}{H_{i}\left(t-1\right)}.$$

Therefore, the maximum likelihood estimator of $p_{i}\left(t\right)$ is defined by
(7)$$\begin{array}{cc} {\hat{p}}_{i}\left(t\right)= & \frac{{\hat{p}}_{i,D}\left(t\right)}{{\hat{p}}_{i,D}\left(t\right)+{\hat{p}}_{i,R}\left(t\right)}\\ = & \frac{n_{i,D}\left(t\right)}{n_{i,D}\left(t\right)+n_{i,R}\left(t\right)}. \end{array}$$

With the passing of time, the severity of the epidemic will change, and the above-mentioned real-time fatality rate can be used to compare the changes in the severity of the epidemic among different groups. Therefore, the following hypothesis
$$H_{0}:\;p_{1}\left(t\right)=p_{2}\left(t\right)=\ldots =p_{k}\left(t\right),\;\;\;H_{1}:\;p_{i}\left(t\right),\;1\leq i\leq k,\;\mathrm{not}\;\mathrm{all}\;\mathrm{equal}$$
is proposed by considering death and recovery as two competing risks. It is easy to see that the above hypothesis is equal to
$$H_{0}:\frac{p_{1,D}\left(t\right)}{p_{1,R}\left(t\right)}=\frac{p_{2,D}\left(t\right)}{p_{2,R}\left(t\right)}=\ldots =\frac{p_{k,D}\left(t\right)}{p_{k,R}\left(t\right)},\;\;\;H_{1}:\frac{p_{i,D}\left(t\right)}{p_{i,R}\left(t\right)},\;1\leq i\leq k,\;\mathrm{not}\;\mathrm{all}\;\mathrm{equal},$$
when the null hypothesis holds true, and we expect that the two competing risks of death and recovery are similar during the observation period. Therefore, we propose the following expression of the sum of squares for factor ($S_{A}$) and the sum of squares for error ($S_{E}$); that is,
(8)$$\begin{array}{cc} S_{A}= & \sum_{i=1}^{k}m_{i}{\left(\frac{\sum_{j=1}^{t}\frac{n_{i,D}\left(j\right)}{n_{i,R}\left(j\right)}}{m_{i}}-\sum_{i=1}^{k}\sum_{j=1}^{t}\frac{\frac{n_{i,D}\left(j\right)}{n_{i,R}\left(j\right)}}{km_{i}}\right)}^{2},\;\;\;f_{A}=k-1;\\ S_{E}= & \sum_{i=1}^{k}\sum_{j=1}^{t}{\left(\frac{n_{i,D}\left(j\right)}{n_{i,R}\left(j\right)}-\sum_{j=1}^{t}\frac{\frac{n_{i,D}\left(j\right)}{n_{i,R}\left(j\right)}}{m_{i}}\right)}^{2},\;\;\;f_{E}=\sum_{i=1}^{k}m_{i}-k. \end{array}$$

The following test statistic is obtained:
(9)$$F=\frac{S_{A}/f_{A}}{S_{E}/f_{E}}=\frac{\mathrm{MSA}}{\mathrm{MSE}}.$$

### 2.2. The Asymptotic Distribution of the Test Statistic

Assume that $P={\left(p_{1},p_{2},\ldots ,p_{n}\right)}^{T}$ is a column vector of an *n*-dimensional normal random variable. Let *A* be an n×n real symmetric matrix; then $P^{T}AP$ is called the quadratic form of *A*. Denote the rank of *A* by *r*. According to the knowledge of linear algebra, it can be seen that the rank of the quadratic $P^{T}AP$ is *r*, for short $\mathrm{rank}\left(A\right)=\mathrm{rank}\left(P^{T}AP\right)=r$.

Suppose that the *n* observation values $p_{i}\left(j\right)$ (i=1,…,k; $j=1,\ldots ,m_{i}$) in one-way analysis of variance are *n* independent random variables and that $p_{i}\left(j\right)∼N\left(\mu_{i},\sigma^{2}\right)$. Set $J_{n}$ is the *n*-order square matrix with all elements 1, $I_{n}$ is the *n*-order unit matrix, and $C=diag(C_{1},C_{2},\ldots ,C_{k})$ is a block diagonal matrix, where $C_{i}=J_{m_{i}}/m_{i}$; that is:$$I_{n}={\left(\begin{array}{ccccc} 1 & \cdots & 0 & \cdots & 0\\ \vdots & \ddots & & & \vdots \\ 0 & & 1 & & 0\\ \vdots & & & \ddots & \vdots \\ 0 & \cdots & 0 & \cdots & 1 \end{array}\right)}_{n\times n},\;\;J_{n}={\left(\begin{array}{ccccc} 1 & \cdots & 1 & \cdots & 1\\ \vdots & \ddots & & & \vdots \\ 1 & & 1 & & 1\\ \vdots & & & \ddots & \vdots \\ 1 & \cdots & 1 & \cdots & 1 \end{array}\right)}_{n\times n}$$
and
$$C={\left(\begin{array}{ccccc} J_{m_{1}}/m_{1} & \cdots & 0 & \cdots & 0\\ \vdots & \ddots & & & \vdots \\ 0 & & J_{m_{i}}/m_{i} & & 0\\ \vdots & & & \ddots & \vdots \\ 0 & \cdots & 0 & \cdots & J_{m_{k}}/m_{k} \end{array}\right)}_{n\times n}.$$

Based on the above assumptions, the following three basic quadratic forms are obtained through calculation:
(10)$$\left\{\begin{array}{cc} P^{T}I_{n}P=\sum_{i=1}^{k}\sum_{j=1}^{m_{i}}p_{i}{\left(j\right)}^{2}; & \\ P^{T}J_{n}P=T^{2}, & \mathrm{where}\;T=\sum_{i=1}^{k}\sum_{j=1}^{m_{i}}p_{i}\left(j\right);\\ P^{T}CP=\sum_{i=1}^{k}T_{i}^{2}/m_{i}, & \;\mathrm{where}\;T_{i}=\sum_{j=1}^{m_{i}}p_{i}\left(j\right). \end{array}\right.$$

Through the quadratic form in (10), Mao et al. [15] proved that $S_{A}$, $S_{E}$, and the sum of squares for total ($S_{T}$) are quadratic forms of normal random vectors, whose ranks are equal to their respective degrees of freedom; therefore, we have
$$\begin{array}{cc} S_{A}=P^{T}A_{1}P, & \;\;A_{1}=C-J_{n}/n,\;\;\mathrm{rank}\left(A_{1}\right)=k-1;\\ S_{E}=P^{T}A_{2}P, & \;\;A_{2}=I_{n}-C,\;\;\mathrm{rank}\left(A_{2}\right)=n-k;\\ S_{T}=S_{A}+S_{E}=P^{T}AP, & \;\;A=I_{n}-J_{n}/n,\;\;\mathrm{rank}\left(A\right)=n-1. \end{array}$$

Under the null hypothesis, we also have
$$\frac{S_{A}}{\sigma^{2}}=\frac{P^{T}A_{1}P}{\sigma^{2}}∼\chi^{2}\left(k-1\right),\;\;\;\;\;\frac{S_{E}}{\sigma^{2}}=\frac{P^{T}A_{2}P}{\sigma^{2}}∼\chi^{2}\left(n-k\right).$$

$S_{A}$ and $S_{E}$ are independent, so
(11)$$F=:\frac{S_{A}/\left(k-1\right)}{S_{E}/\left(n-k\right)}=\frac{\mathrm{MSA}}{\mathrm{MSE}}∼F\left(k-1,n-k\right).\;$$

Here, *F* is the probability distribution of the test statistic required for one-way analysis of variance. Then, for the given significance level α>0, it can be determined that the rejection region is
$$W=\left\{F>F_{1-\alpha }\left(k-1,n-k\right)\right\}.$$

Since the data used in the analysis of variance need to satisfy the assumptions of normality, homogeneity of variance and independence are needed; however, since the data in this study do not satisfy these assumptions, we will use the permutation test for the variance analysis, which is introduced in detail in the next section.

## 3. Permutation Test

When the data do not satisfy the premise assumptions of normality, homogeneity of variance, or independence in the variance analysis, the variance analysis cannot be conducted [16]. In this case, statistical methods based on randomization and resampling can be used for testing. We use a widely applicable statistical method from the idea of randomization, which is called a permutation test. Xue et al. [17] pointed out that the permutation test has obvious advantages over traditional statistical methods because it does not need to make presuppositions about the distribution and is conducted solely based on the information contained in the observed samples.

The core problem to be solved by using the permutation test for variance analysis is how to estimate the probability distribution of the test statistic under the condition that the null hypothesis is established. The basic thought is that if the null hypothesis that states that there is no significant difference in the mean of the sample data from *k* groups is true, then the null hypothesis cannot be overturned by calculating the observation value (denoted as $F_{0}$) of its test statistics based on the observed *k* independent samples. At this time, if *k* samples are mixed and then randomly divided into *k* new samples, also known as permutation samples, the observed values of the test statistics are calculated again and tested, and the same inferred conclusion will be obtained.

Assuming that it is necessary to test whether there is any difference between the effect among *k* groups, then set a total of $n=\sum_{i=1}^{k}m_{i}$ (i=1,…,k) experimental units, and divide these experimental units into *k* parts. Each part contains $m_{i}$ units, and *k* different treatments are applied separately. The test results obtained from each treatment are recorded as $Y_{i}=\left(y_{i1},\ldots ,y_{ij},y_{im_{i}}\right)$ ($i=1,\ldots ,k;j=1,\ldots ,m_{i}$), and the effect value is $U_{i}=\left(u_{i1},\ldots ,u_{ij},u_{im_{i}}\right)$; then the assumption to be considered is
$$H_{0}:\;U_{1}=U_{2}=\ldots =U_{k}.$$
when assigning experimental units based on the principle of randomization. By arbitrarily lining up *n* experimental units into a row and applying *k* different treatments to $m_{i}$ units in turn, the data $y_{ij}$ are randomly extracted from the effect $u_{ij}$. If the following test statistic is used
(12)$$F^{\prime }=\frac{\mathrm{MSA}}{\mathrm{MSE}}=\frac{\sum_{i=1}^{k}m_{i}{\left({\bar{y}}_{i}-\bar{y}\right)}^{2}/\left(k-1\right)}{\sum_{i=1}^{k}\sum_{j=1}^{m_{i}}{\left(y_{ij}-{\bar{y}}_{i}\right)}^{2}\left(\sum_{i=1}^{k}m_{i}-k\right)},\;$$
then $F^{\prime }$ is evaluated with the probability of $\frac{1}{n!}$ every time; $U_{i}=\left(u_{i1},\ldots ,u_{ij},u_{im_{i}}\right)$ is arranged in a row and divided into *k* parts, where the value of each part is $Y_{i}$; and the value of $F^{\prime }$ can be calculated according to (12). By repeating this process, n! values of $F^{\prime }$ can be obtained, denoted as $F_{1}^{\prime },\ldots ,F_{n_{i}}^{\prime }$. Arrange all $F_{1}^{\prime },\ldots ,F_{n_{i}}^{\prime }$ into $F_{\left(1\right)}^{\prime }\leq F_{\left(2\right)}^{\prime }\leq \ldots \leq F_{\left(n!\right)}^{\prime }$ according to size, and select an appropriate $F_{\left(g\right)}^{\prime }$ (g=1,…,n!) according to significance level α.

When $F_{0}>F_{\left(g\right)}^{\prime }$, reject the null hypothesis, believing that there is a significant difference in the effect among *k* groups. The following is the rejection region of the permutation test:(13)$$W^{\prime }=\left\{F_{0}>F_{1-\alpha }^{\prime }\right\}.\;$$

Equation (Equation 13) uses the quantile of the permutation distribution as the critical value to determine the rejection region. It can be seen that both the permutation method and the parameter method need to calculate the test statistic. The difference is that the permutation method does not compare the test statistic with the theoretical distribution but rather compares the test statistic with the empirical distribution obtained by the permutation sample and judges whether there is enough evidence to reject the null hypothesis according to the extremeness of the statistical value.

## 4. Simulation

A large-scale simulation is carried out to evaluate the performance of the proposed approach, generating simulated data for daily confirmed cases, with deaths and recoveries unknown. This simulates the real-world epidemiological data in which only aggregated counts are reported during an epidemic outbreak. We assume that the observation period is 50 days (T=50), and it needs to take into account how real-time fatality rates changes over time in practice. Therefore, we assume that, during the 50-day observation period, when $0<t\leq t_{0}$, the daily number of confirmed cases follows the Poisson distribution of parameter $\lambda_{0}$, and when $t_{0}<t\leq 50$, the daily number of confirmed cases follows the Poisson distribution of parameter $\lambda_{1}$. In addition, the simulation scenarios are based on the preassigned death probability and recovery probabilities on day *t* for group *i*. The daily number of deaths and recoveries are generated according to the multinomial setting in (1), and the real-time fatality rates are further calculated according to the generated simulation data. Finally, the permutation test is used for variance analysis to determine the differences in disease severity among *k* groups.

Under $H_{0}$, the performance of the proposed test is evaluated by considering various scenarios in which the real-time fatality rates are equal among three groups (group A, group B, group C). From the simulation results in Table 1, we can see that the analysis of variance using the permutation test can sensitively capture the differences in real-time fatality rate among different groups, thus rejecting the null hypothesis of equal real-time fatality rate.

Figure 1 shows the multiple comparison tests of the first four scenarios at the confidence level α=0.05. The upper left corner shows that the confidence interval of the mean difference of each group in scenario 1 does not contain the 0 point. Therefore, we can judge that there is a significant difference in the real-time fatality rate among the three groups in scenario 1. The upper right corner shows the confidence interval of the mean difference of each group in scenario 2, and only the confidence interval of the mean difference between group A and group B contains the 0 point, which means that there is no significant difference in the real-time fatality rate between group A and group B. The lower left corner and the lower right corner respectively show the confidence interval of the mean difference of each group in scenarios 3 and 4. In these two figures, the confidence interval of mean difference of group B and group C contains the 0 point; that is, there is no significant difference in the real-time fatality rate between group B and group C under scenarios 3 and 4.

## 5. Empirical Analysis

COVID-19 is the most widespread global pandemic to hit humanity in the past century. It is a serious crisis for the whole world, posing a major threat to human life and health [18,19]. The first case of COVID-19 was found in Wuhan, Hubei Province, China, in December 2019. At present, China’s National Health Commission has renamed it as the Novel Coronavirus Infection. Kraemer et al. [20] found that, despite restrictions on mobility imposed in Wuhan on 23 January 2020, it still could not stop the rapid spread of COVID-19 to other cities. According to the statistics of the National Health Commission [21], as of 24:00 on 29 February 2020, a total of 79,824 confirmed cases had been reported in mainland China, including 2870 deaths; 66,907 confirmed cases and 2761 deaths had been reported in Hubei Province; and 49,122 confirmed cases and 2195 deaths had been reported in Wuhan. At first glance, the mortality in mainland China is about 3.60%, the mortality in Hubei Province is about 4.13%, and the mortality in Wuhan is about 4.47%. However, it is difficult to estimate the mortality of an epidemic, especially for the outbreak of COVID-19. Medical staff all over China have to make every effort to deal with this extremely infectious and terrible disease. Next, the proposed method is applied to analyze the differences in disease severity among different regions over time.

## 6. Data Description

Considering that the intervention measures implemented during the epidemic and availability of medical resources may have a potential impact on the fatality rate in the initial stage of COVID-19 in China, we have divided mainland China into three different clusters, namely mainland China excluding Hubei Province, Hubei Province excluding Wuhan, and Wuhan, and we use the real-time fatality rate to analyze the differences in disease severity among different clusters. The daily number of confirmed cases, daily deaths, and daily recoveries for each cluster from the official website of the National Health Commission (http://www.nhc.gov.cn, accessed on 11 June 2020) were extracted and summarized from 31 January 2020 to 30 March 2020.

Figure 2 shows the real-time fatality rate estimator of the three independent clusters throughout the observation period. We can see that, in the initial stage of the COVID-19 epidemic, there are obvious differences in disease severity among different regions in mainland China. With the passage of time, the difference in the severity of the disease gradually decreases. In this regard, we provide some explanations to describe the observed phenomena.

In the initial stage of the epidemic, most of the confirmed cases were mainly concentrated in Hubei Province. Given the large number of patients and the lack of clinical experience with the disease, the local medical systems were overwhelmed. In Wuhan in particular, many patients were not treated in time, causing the highest mortality there, followed by Hubei Province (excluding Wuhan). On the other hand, the implementation of strict lockdown measures in Wuhan curbed the spread of the disease to a certain extent, providing valuable preparation time for other regions to prevent the epidemic. In the face of the unknown, sudden, and fierce natural disasters of the epidemic, the Chinese government decisively launched a battle of epidemic prevention and control and mobilized all necessary resources throughout the country to support virus control in Hubei Province and Wuhan. Leishenshan Hospital and Huoshenshan Hospital, which were used to focus on the treatment of COVID-19 patients, had 1500 beds and 100 beds, respectively. These two hospitals took only about ten days to realize the whole process, from the completion of the design plan to delivery. Cities in Hubei Province received paring assistance from other provinces [9]. As of 14 February 2020, more than 25,000 medical professionals rushed to Wuhan. In addition, Wuhan successfully established 16 Fangcang shelter hospitals, which eased the great pressure on the medical system [22]. Meanwhile, more than 40 designated hospitals were set up in Wuhan, mainly to treat severe COVID-19 patients [23]. During the observation period, the real-time fatality rate in mainland China (excluding Hubei Province) was basically stable at a low level. With the continuous alleviation of the pressure of medical resources in Wuhan and Hubei Province, their real-time fatality rate generally showed a continuous downward trend. Finally, by late February 2020, the real-time fatality rates in the three regions were no longer significantly different, and the level remained relatively low in March 2020.

### Empirical Analysis Result

In this section, the proposed permutation test is applied to the analysis of variance to test the degree of difference in real-time fatality rates over time among different clusters. The unit is a day, which means there are T=60 time intervals, and the 0th day (T=0) is 31 January 2020. A three-sample test is conducted on $H_{0}$:$$H_{0}:\;p_{\mathrm{China}}\left(t\right)=p_{\mathrm{Hubei}}\left(t\right)=p_{\mathrm{Wuhan}}\left(t\right),\;\;\;\;\;0<t\leq T$$
and we set the overall significance level α to 0.05.

Specifically, during the 60-day observation period, the test statistic $F_{0}$ of the original sample is first calculated. Then, we mix the three original samples, randomly assign the original samples into three groups to obtain the permutation samples, calculate the test statistic $F_{i}$ of the permutation samples, and repeat this n! times. Finally, $F_{i}$ is arranged in order, making the distribution of $F_{i}$ the empirical distribution. The decision is made by comparing $F_{0}$ with the empirical distribution. According to Table 2, there are generally significant differences in the real-time fatality rates of the three regions during the 60-day observation period. The multiple comparison test results in Figure 3 show that the confidence interval of the mean difference in real-time fatality rate between two or two of the three regions does not contain the 0 point, which shows significant difference in disease severity among the three regions during the whole observation period. According to Table 3, we can see that the real-time fatality rate in Hubei is 0.0532 higher than in China on average, while the real-time fatality rate in Wuhan is 0.1010 higher than in China on average, and 0.0478 higher than in Hubei.

Further, in order to measure the effectiveness of the intervention measures implemented by the Chinese government and the impact of improved medical resources on disease severity, we conduct a temporal examination of the real-time fatality rates in Hubei and Wuhan before and after assistance. First, the proposed method is performed for the period from 31 January 2020 to 26 February 2020, when T=27. According to the results in Figure 4, before the aid to Hubei and Wuhan and at the initial stage of the aid, the confidence interval of the mean difference of the real-time death rate between the three regions does not contain the 0 point, suggesting that there are still significant differences in the real-time fatality rate between the three regions.

In the late assistance period, the same inspection procedure is carried out from 27 February 2020 to 30 March 2020, when T=33. As shown in Figure 5, during this period, the confidence interval of the mean difference in the real-time fatality rate in the three regions contains the 0 point, suggesting that there is no significant difference in the real-time fatality rate in the three regions. It can be considered that the accessibility of medical resources in different regions is roughly the same, and the intervention policies implemented by the Chinese government have effectively alleviated the disease severity. The result shown in Figure 5 further confirms similar real-time fatality rates in the three regions from late February 2020 to late March 2020.

## 7. Conclusions

In this paper, a non-parametric test procedure is proposed. The non-parametric test method generally does not involve population parameters, and its assumptions are much less than those of a parametric test. Therefore, nonparametric tests are used to compare differences in real-time fatality rates among independent groups during epidemic outbreaks. Since implementing effective interventions can inhibit the spread of the disease to some extent, thereby reducing the disease severity and saving more lives, our approach can validate the effectiveness of interventions. The asymptotic distribution of the test statistic under the null hypothesis allows analysis of variance to be performed using the permutation test, and the null hypothesis can be rejected when there are significant differences in disease severity among groups. At the same time, our simulations show that the proposed test can accurately identify inter-group differences in disease severity for various scenarios. The test procedure proposed in this paper is applied to the data of the initial stage of the COVID-19 epidemic in mainland China to test the difference in the severity of the disease among the three independent clusters. The results show that there is a significant difference in the real-time fatality rate of the three clusters during the whole observation period. Through the discussion of time segments, we found that with the implementation of effective intervention measures and the improvement in medical resources, there is no significant difference in the real-time fatality rate of the three clusters in the later stage of observation. We can see that the severity of COVID-19 in mainland China may be related to the implementation of interventions and the availability of medical resources, which reflects the important role of effective interventions and medical resources in reducing the real-time fatality rate.

Our method can be applied to the case of *k* samples. In essence, this allows researchers to study more clinical questions in concrete real-life situations, such as investigating the extent of differences in the real-time fatality rate between multiple sex groups or age groups, which may help study other factors that influence the real-time fatality rate of diseases. It is important to note that different treatment regimens are used clinically for different groups, and investigating the extent to which mortality varies between treatment groups can help healthcare providers determine the most effective treatment regimens. The method proposed in this paper can be one of the important methods to evaluate the efficacy of different treatment regimens. The test procedure is based on the information contained in the observed sample and is not limited by the population distribution. Surveillance data from authorities during outbreaks is always incomplete, with some data such as gender and age being difficult to obtain, especially in areas with low awareness of protection and inadequate healthcare systems. Despite the post-pandemic period, detailed data on the COVID-19 are still difficult to obtain, and for most regions, only daily confirmed cases, deaths, and recoveries are recorded and aggregated. It is most important for the health department to use the simplest data structures to gain a deeper understanding of the disease so that rapid interventions can be taken to curb the spread of the disease early and reduce mortality.

## Figures and Tables

**Figure 1 entropy-25-01028-f001:**
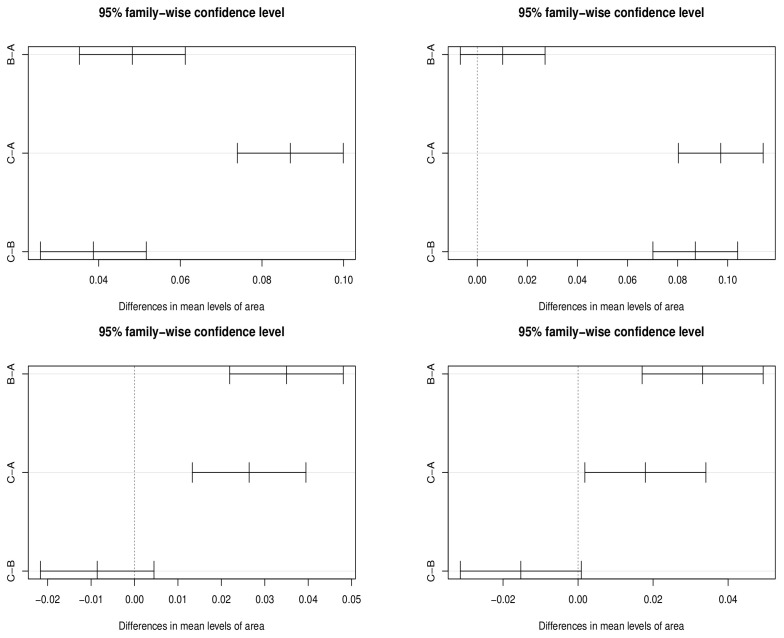
Multiple comparison tests in the first four simulated scenarios.

**Figure 2 entropy-25-01028-f002:**
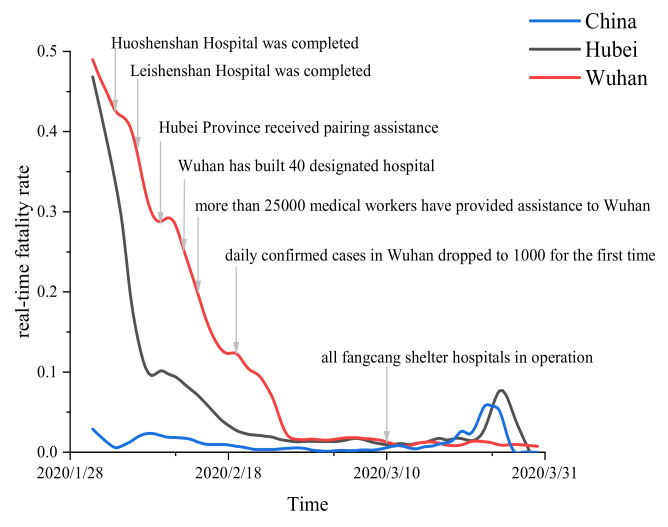
Changes in real-time fatality rate in different clusters during the observation period.

**Figure 3 entropy-25-01028-f003:**
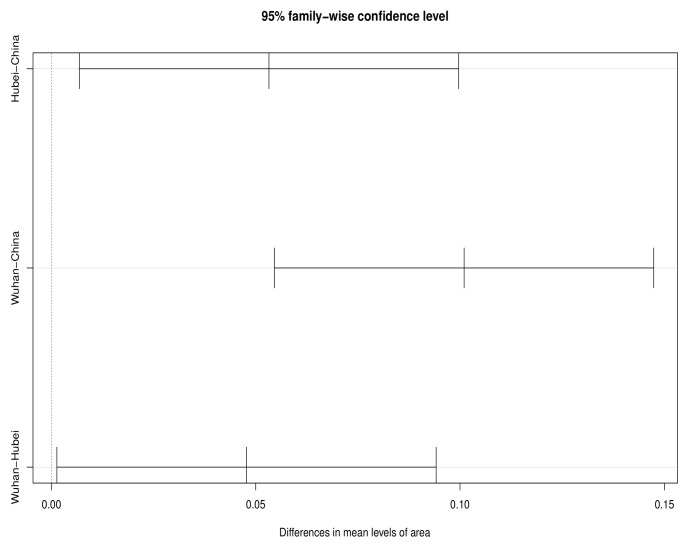
Confidence intervals for mean differences in real-time fatality rate across the observation period.

**Figure 4 entropy-25-01028-f004:**
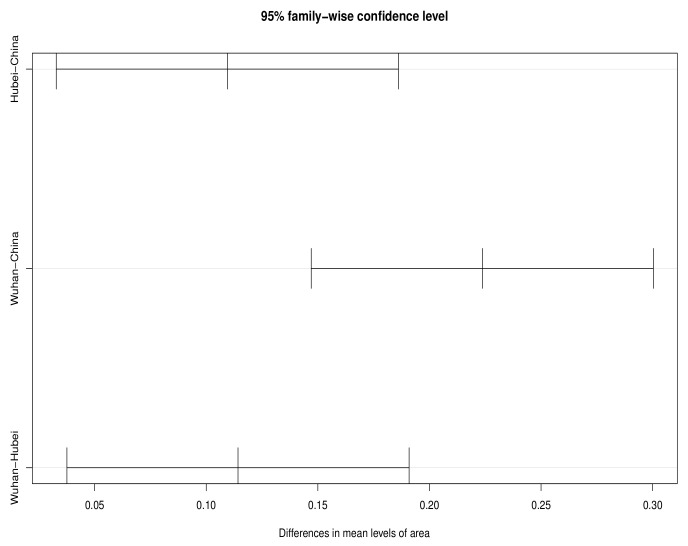
Confidence intervals for mean differences in real-time fatality rate among three regions before and at the beginning of the assistance.

**Figure 5 entropy-25-01028-f005:**
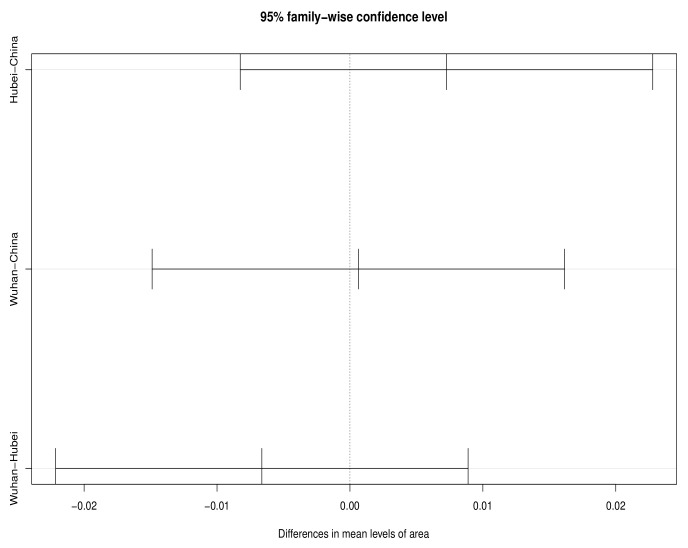
Confidence intervals for mean differences in real-time fatality rate among three regions in the post-assistance period.

**Table 1 entropy-25-01028-t001:** Simulation results.

Scenario	$p_{1,D}\left(t\right)$	$p_{1,R}\left(t\right)$	$p_{2,D}\left(t\right)$	$p_{2,R}\left(t\right)$	$p_{3,D}\left(t\right)$	$p_{3,R}\left(t\right)$	*p*-Value	Multiple Comparison
1	0.01	0.9	0.06	0.9	0.1	0.89	≪0.00	*√*
2	0.01	0.9	0.02	0.89	0.1	0.8	≪0.00	C-A,C-B
3	0.01	0.74	0.05	0.89	0.03	0.75	≪0.00	B-A,C-A
4	0.01	0.74	0.04	0.8	0.02	0.75	≪0.00	B-A,C-A
5	0.02	0.80	0.02	0.8	0.03	0.8	0.0316	C-B
6	0.02	0.80	0.02	0.85	0.03	0.85	0.1312	−
7	0.02	0.80	0.04	0.8	0.02	0.9	≪0.00	B-A,C-B
8	0.02	0.60	0.06	0.74	0.02	0.82	≪0.00	B-A,C-B
9	0.05	0.7	0.02	0.63	0.02	0.6	≪0.00	B-A,C-B
10	0.05	0.65	0.05	0.67	0.04	0.7	0.672	−
11	0.05	0.6	0.1	0.74	0.04	0.7	≪0.00	B-A,C-B
12	0.05	0.82	0.03	0.74	0.25	0.74	≪0.00	*√*
13	0.14	0.78	0.07	0.74	0.11	≪0.00	B-A,C-B	
14	0.14	0.75	0.20	0.60	0.27	0.6	≪0.00	*√*
15	0.14	0.7	0.17	0.75	0.22	0.74	≪0.00	C-A,C-B
16	0.14	0.7	0.08	0.75	0.27	0.61	≪0.00	*√*

*√*: there is a difference between any two in the groups ; −: there is no difference among all groups.

**Table 2 entropy-25-01028-t002:** Permutation test variance analysis.

Source	DF	SQ	MSQ	F	*p*-Value
Region	2	0.30638	0.15319	13.25173	<0.00000
Residuals	177	2.04664	0.01156		
Total	179	2.35302			

DF = degrees of freedom; SQ = sum of squares; MSQ = sum of mean squares; F = the test statistic.

**Table 3 entropy-25-01028-t003:** Multiple comparison test.

Region	Diff	Lwr	Upr	*p*-Value
Hubei–China	0.0532	0.0068	0.0997	0.0200
Hubei–China	0.1010	0.0546	0.1474	0.0000
Wuhan–Hubei	0.0478	0.0013	0.0942	0.0421

Diff = mean difference; Lwr = lower confidence limit; Upr = upper confidence limit.

## Data Availability

The datasets on the daily confirmed cases, deaths and recoveries for each cluster from the official website of the National Health Commission (http://www.nhc.gov.cn, accessed on 11 October 2020).

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
