# Peer review of "A New Analysis of Real-Time Fatality Rate in the Initial Stage of COVID-19"

_entropy, 2023, doi:10.3390/e25071028_

Round 1

Reviewer 1 Report

The paper adds no relevant results on the subject considered or any interesting application or improvement concerning the methods employed for the analysis. As a side comment, it is not well written also in the section describing the methodology employed. The formula shown should be thoroughly revised. Further and more relevant references could have been added.

The English should be thoroughly revised since some sentences are difficult to be understood and some others are incomplete or with errors. 

Author Response

According to the report of reviewer, we make a corresponding modification. See the cover letter

Reviewer 2 Report

In this paper, a test procedure of non-parametric method is proposed by authors. They also presented several scenario simulations for COVID-19 epidemic in mainland China mortality cases which can be used to validate the effectiveness of interventions. 

The results showed that there is a significant difference in the real-time fatality rate of the three clusters during the whole observation period.

To the best of my knowledge, this work is new, insightful, and will interest the readers of this journal and I am of the opinion that after minor revision, it can be accepted for publication.

Authors should read thoroughly and correct all typographical errors in the manuscript.

Authors should placed the tables in the manuscript properly. I suspect this is due to the latex command used by authors. It was difficult to understand the tables as some parts are cut off.

Author Response

According to the report, we make a modification. See the cover letter

Reviewer 3 Report

Dear authors, please explain why you have excluded some major cities? This should be addressed and its possible influences on the results should be discussed.

Please have a final English language edit

Author Response

We make a modification in the manuscript. See the cover letter.
